# Modeling functional cell types in spike train data

**Daniel N. Zdeblick**[1]*, **Eric T. Shea-Brown**[2,3], **Daniela M. Witten**[4], **Michael A. Buice**[2,3]

**1** Department of Electrical and Computer Engineering, University of Washington, Seattle, Washington, United States of America, **2** Department of Applied Math, University of Washington, Seattle, Washington, United States of America, **3** MindScope Program, Allen Institute, Seattle, Washington, United States of America, **4** Department of Statistics and Biostatistics, University of Washington, Seattle, Washington, United States of America

* zdeblick@uw.edu

**Data Availability Statement:** All code used in this study is available at https://github.com/zdeblick/ClusteredGLMs.

**Funding:** This work was supported by the National Institutes of Health (R01DA047869 to DW;

## Abstract

A major goal of computational neuroscience is to build accurate models of the activity of neurons that can be used to interpret their function in circuits. Here, we explore using *functional cell types* to refine single-cell models by grouping them into functionally relevant classes. Formally, we define a hierarchical generative model for cell types, single-cell parameters, and neural responses, and then derive an expectation-maximization algorithm with variational inference that maximizes the likelihood of the neural recordings. We apply this "simultaneous" method to estimate cell types and fit single-cell models from simulated data, and find that it accurately recovers the ground truth parameters. We then apply our approach to *in vitro* neural recordings from neurons in mouse primary visual cortex, and find that it yields improved prediction of single-cell activity. We demonstrate that the discovered cell-type clusters are well separated and generalizable, and thus amenable to interpretation. We then compare discovered cluster memberships with locational, morphological, and transcriptomic data. Our findings reveal the potential to improve models of neural responses by explicitly allowing for shared functional properties across neurons.

## Author summary

Computational neuroscience has its roots in fitting and interpreting predictive models of the activity of individual neurons. In recent years, more attention has focused on models of how ensembles of neurons work together to perform computations. However, fitting these more complex models to data is challenging, limiting our ability to use them to understand real neural systems. One idea that has improved our understanding of populations of neurons is *cell types*, where neurons of the same type have similar properties. While the idea of cell types is old, recent work has focused on functional cell types, where the properties of interest are derived from fitted predictive models of the activity of single neurons. In this work, we develop a method that *simultaneously* fits a predictive model of each neuron's activity and groups neurons into functional cell types. Compared to existing

R01EB026908 to DW and MB) and a Simons Investigator Award in Mathematical Modeling of Living Systems (to DW). The funders had no role in study design, data collection and analysis, decision to publish, or preparation of the manuscript.

**Competing interests:** The authors have declared that no competing interests exist.

techniques, this method produces more accurate models of single-cell neural activity and better groupings of neurons into types. This method can thus contribute the use of cell types in better understanding the components of neural systems based on our increasingly rich observations of their functional responses.

## 1 Introduction

A primary goal of computational neuroscience is to formulate simplifying assumptions on neural structure and activity that allow for models that are both tractable to fit to data and useful for understanding neural systems. One such simplifying assumption that is gaining traction is that all neurons in the brain belong to a one of a finite number of *cell types*, and that neurons of the same type have similar properties. Many studies have sought to cluster neurons into putative cell types according to properties of their morphology [1], gene expression [2–4], electrophysiology [5, 6], and connectivity [7].

We are interested here in *functional cell types* that group neurons with similar properties of functional output, which we consider to be their spiking response. Clearly, there is a strong relationship between this notion and cell types defined on more specific properties of electrophysiological responses, and potentially transcriptomic and morphological types as well. However, this functional view of cell types only imputes value to differences between neurons that are useful in predicting their spiking responses to stimuli. The work of Teeter et al. [6] took an important step towards identifying functional cell types from electrophysiology data with a sequential approach. Specifically, these authors fit *functional models of single-neuron responses*, and then clustered the resulting parameter estimates. By using functional model parameters as features to be clustered, these authors explicitly relate functional cell types to prediction of neural responses. Like the authors of that work, we believe that this relationship is crucial to identifying cell types that will help us understand the brain's function.

The present work has two primary goals: to use the idea of functional cell types to improve parameter estimates for predictive models of individual neural responses, and to discover the best possible grouping of cells into types. To meet these twin goals, we develop an approach that *simultaneously* estimates single-cell model parameters and cell types (clusters of those parameters). All of the previous studies on discovering cell types cited above, except [7], take a *sequential approach* to defining cell types: they extract features of interest from each neuron's data, and then cluster these features using an unsupervised clustering algorithm. Unlike these sequential approaches, our method allows estimates of each individual neuron's parameters to "borrow strength" from other neurons' data. As we will show, our simultaneous method leads to more accurate single-cell models and cell-type clusterings than a matched sequential approach. Even in real-world situations where there are no "ground truth" cell types, the simultaneous method yields improved prediction of single-cell responses and clusters that are more robust to the exclusion of different neurons from training, providing validation for this approach to discovering cell types.

In greater detail, we define a hierarchical generative model of functional cell types, single-cell parameters, and neural responses. In our model, each neuron belongs to one of several possible (unknown) functional cell types; the distribution of parameters for that neuron's response model is governed by the (unknown) cell type to which it belongs. This is a mixture model in which each sample is the entire spike train from a single neuron, and the distribution of this sample, conditional on its cluster membership, requires marginalizing out parameters of the response model. To fit this hierarchical model, we adapt the expectation-maximization

(EM) algorithm [8] to simultaneously estimate the parameters for each neuron's response model and the functional cell types, in order to provide a good fit to the spiking response of the recorded neurons.

We apply our method to simulated data, as well as to the Allen Cell Types Database collected by the Allen Institute for Brain Science [9] (also used in [6]). We verify that our approach accurately recovers the ground truth model underlying a simulated dataset, and demonstrate that it discovers robust and interpretable cell types that improve prediction of neural recordings. In particular, we show that the benefits of applying our method to predict neural activity depend on the training data in a way that is consistent with the notion of "borrowing strength"; the improvement is greatest when the training dataset includes more neurons.

For comparison with the simultaneous method, we use a version of the sequential method with the same model structure, but where single-cell parameters are first fit, then clustered into cell types. This comparison reveals the potential improvement of borrowing strength between neurons.

## 2 Methods

We begin by formalizing the goals of this work, and then detail two approaches to meet them: a "sequential method," based on approaches pervading existing cell-types research, and a "simultaneous method," our proposed improvement. We then adapt the EM algorithm for our model in order to estimate the parameters of the model from data, and use the Bayesian information criterion (BIC) to select hyperparameters. All code used to carry out the analyses in this paper is available at https://github.com/zdeblick/ClusteredGLMs.

Throughout, we use bold symbols (e.g. $\mathbf{x}_i$) to denote vectors where the $t$th element is denoted e.g. $x_i(t)$, and capital Latin letters to denote natural number constants ($N$, $K$, $T_i$). We use $f(\mathbf{z}; \boldsymbol{\mu}_z, \Sigma_z)$ to denote the probability density function of a multivariate Gaussian distribution, and a hat (e.g. $\hat{\boldsymbol{\beta}}_i$) to denote an estimate.

### 2.1 Goals

In this work we seek a clustering of neurons into cell types that best explains their functional (spiking) responses. We consider a dataset of spiking responses to stimuli $x_i(t)$, $y_i(t)$, $t \in \{1, \ldots, T_i\}$, $i \in \{1, \ldots, N\}$, where $N$ is the number of neurons, $y_i(t)$ is the number of spikes that the $i^{th}$ neuron fires in the $t^{th}$ time bin, $x_i(t)$ is the value of the stimulus to that neuron in that time bin, and $T_i$ is the duration of that neuron's recording (in time bins). Our goals can then be formalized as estimating two quantities from our dataset:

1. For each neuron, $i, \ldots, N$, parameters $\hat{\boldsymbol{\beta}}_i$ of a predictive model for the $i$th neuron's response, $P(\mathbf{y}_i|\mathbf{x}_i, \hat{\boldsymbol{\beta}}_i)$.

2. Functional cell-type assignment of each neuron into one of $K$ types, $\hat{k}_i \in \{1, \ldots, K\}$, that best capture the distribution of $\hat{\boldsymbol{\beta}}$ across all neurons.

We compare two approaches for achieving these goals: a "sequential method" of first estimating $\hat{\boldsymbol{\beta}}_1, \ldots, \hat{\boldsymbol{\beta}}_N$, and then clustering these point estimates into cell-type assignments; and a "simultaneous method" that uses an expectation-maximization algorithm to estimate both $\hat{\boldsymbol{\beta}}_i$ and $\hat{k}_i$ in tandem. The sequential method is motivated by the approach of Teeter et al. [6], although implementation details differ.

## 2.2 Sequential method

The sequential method defines functional cell types in two steps. In the first step, the data $\mathbf{x}_i$, $\mathbf{y}_i$ for the $i$th neuron are fit with a single-cell model, which we describe below, that is parameterized by some vector of parameters $\boldsymbol{\beta}_i$. In the second step, the estimated parameters $\hat{\boldsymbol{\beta}}_i$ are clustered with a Gaussian mixture model. See Algorithm 1.

**2.2.1 Single-cell model.** The goal of the single-cell model is to predict $\mathbf{y}_i$ from $\mathbf{x}_i$, using some learned vector of parameters $\boldsymbol{\beta}_i$. We assume the conditional independence of time bins in order to decompose this probability as follows:

$$P_{SC}(\mathbf{y}_i|\mathbf{x}_i;\boldsymbol{\beta}_i) = \prod_{t=1}^{T_i} P_{SC}(y_i(t)|y_i(1),...,y_i(t-1),x_i(1),...,x_i(t);\boldsymbol{\beta}_i). \tag{1}$$

Here, we use $P_{SC}$ to denote the probability distribution of spiking for a single-cell. While there are many options for models of single-cell spiking that parameterize this probability, in this work we use the generalized linear model (GLM) [10], illustrated in Fig 1A. Specifically, we fit a Poisson GLM on a transformed set of covariates:

$$y_i(t) \sim Poisson\left(\exp\left[\sum_{\tau=0}^{T^{\text{stim}}-1}\beta_i^{\text{stim}}(\tau+1)\tilde{x}_i(t-\tau d^{\text{stim}}) + \sum_{\tau=1}^{T^{\text{self}}}\beta_i^{\text{self}}(\tau)y_i(t-\tau) + \beta_i^0\right]\right). \tag{2}$$

In (2):

- $\boldsymbol{\beta}_i^{\text{stim}} \in \mathbb{R}^{T^{\text{stim}}}$ is called the *stimulus filter* for the $i$th neuron, as it filters the stimulus provided to that neuron.

- $\tilde{x}_i(t) = \sum_{s=0}^{d^{\text{stim}}-1} x_i(t-s)$ is the pre-filtered stimulus, using a rectangular filter of length $d^{\text{stim}}$. Accordingly, note that the stimulus filter's coefficients are effectively spaced $d^{\text{stim}}$ time bins apart. The stimuli we will consider vary much more slowly than the timescale of spiking, and this feature reduces the dimensionality of $\boldsymbol{\beta}_i^{\text{stim}}$ to combat overfitting by effectively downsampling the stimulus. As such, we refer to $d^{\text{stim}}$ as the stimulus downsampling factor.

- $\boldsymbol{\beta}_i^{\text{self}} \in \mathbb{R}^{T^{\text{self}}}$ is called the *self-interaction filter* for the $i$th neuron, as it filters that neuron's own history of recent spiking activity.

- $\beta_i^0$ is the offset term for the $i$th neuron.

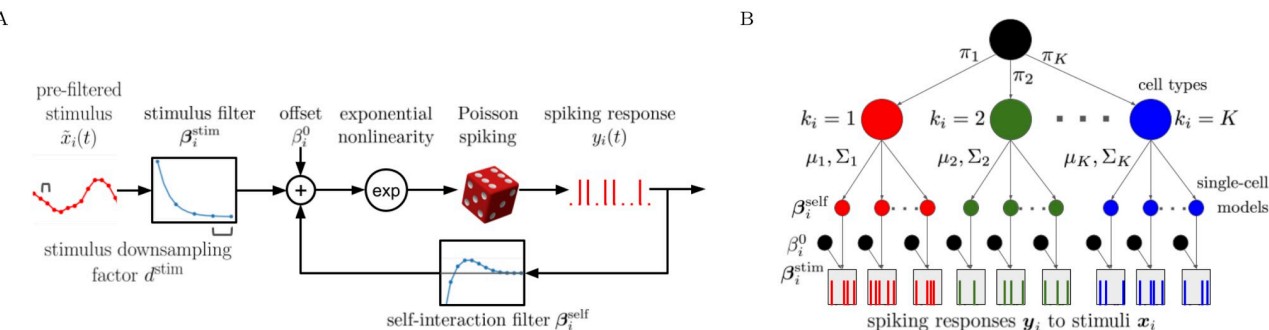

**Fig 1.** A: The Poisson GLM (2) used to model the spiking response of a single neuron. B: The generative model (11) for the simultaneous method.

- Collectively, $\boldsymbol{\beta}_i \equiv [\boldsymbol{\beta}_i^{\text{stim}}, \boldsymbol{\beta}_i^{\text{self}}, \beta_i^0]^\top$, yielding a total of $\dim(\boldsymbol{\beta}_i) = T^{\text{stim}} + T^{\text{self}} + 1$ parameters of the single-cell model. We use these superscripts with many symbols throughout this document, clarifying the component(s) of $\boldsymbol{\beta}_i$ with which they are associated (the absence of a superscript corresponds to all of $\boldsymbol{\beta}_i$).

- We take $x_i(\tau) = y_i(\tau) = 0$ for $\tau < 1$. That is, we zero-pad our data.

When we maximize the corresponding log-likelihood with respect to $\boldsymbol{\beta}_i$, we include $\ell_2$ regularization on all parameters except the offset:

$$\hat{\boldsymbol{\beta}}_i(\lambda^{\text{stim}}, \lambda^{\text{self}}) = \arg\max_{\boldsymbol{\beta}} \left\{ \frac{1}{T_i^{\text{train}}} \log P_{SC}(\mathbf{y}_i^{\text{train}} | \mathbf{x}_i^{\text{train}}; \boldsymbol{\beta}) - \frac{1}{2}\lambda^{\text{stim}} ||\boldsymbol{\beta}^{\text{stim}}||_2^2 - \frac{1}{2}\lambda^{\text{self}} ||\boldsymbol{\beta}^{\text{self}}||_2^2 \right\}. \quad (3)$$

In (3), $||\cdot||_2$ denotes the $\ell_2$-norm of a vector, and the superscript "train" indicates that we are using a training set to fit the parameters $\boldsymbol{\beta}_i$. This optimization problem is convex. We solve it with the trust-region Newton-conjugate-gradient algorithm (`trust-ncg` in the `scikit-learn minimize` function [11]).

To select the regularization hyperparameters $\lambda^{\text{stim}}$ and $\lambda^{\text{self}}$, we use cross-validation. That is, each neuron's data is partitioned into $L$ equally-sized bins of adjacent timepoints, $(\boldsymbol{x}_i^1, ..., \boldsymbol{x}_i^L)$ and $(\boldsymbol{y}_i^1, ..., \boldsymbol{y}_i^L)$. For a logarithmically spaced grid of choices of $\lambda^{\text{stim}} \in [10^{-7}, 1]$, $\lambda^{\text{self}} \in [10^{-7}, 1]$, we then compute the cross-validated log-likelihood, averaged over all data points from all $N$ neurons:

$$VALL(\lambda^{\text{stim}}, \lambda^{\text{self}}) = \sum_{i=1}^{N} \frac{1}{NT_i} \sum_{l=1}^{L} \log P_{SC}(\boldsymbol{y}_i^l | \mathbf{x}_i^l; \hat{\boldsymbol{\beta}}_i(\lambda^{\text{stim}}, \lambda^{\text{self}})). \quad (4)$$

In (4), $\hat{\boldsymbol{\beta}}_i(\lambda^{\text{stim}}, \lambda^{\text{self}})$ is computed using (3), where the other partitions of the data $((\boldsymbol{x}_i^1, ..., \boldsymbol{x}_i^{l-1}, \boldsymbol{x}_i^{l+1}, ..., \boldsymbol{x}_i^L)$ and $(\boldsymbol{y}_i^1, ..., \boldsymbol{y}_i^{l-1}, \boldsymbol{y}_i^{l+1}, ..., \boldsymbol{y}_i^L))$ are used for training.

The $\lambda^{\text{stim}}, \lambda^{\text{self}}$ that maximize this quantity are selected and used to fit $\hat{\boldsymbol{\beta}}_i$ to all of the data used for cross-validation (i.e. not including test data).

We choose a Poisson GLM in (2) because its log likelihood is convex with respect to its parameters $\boldsymbol{\beta}_i$, simplifying estimation. Additionally, the GLM has been widely used to model single-cell responses (see [12] for a review), and is able to produce a wide variety of spiking dynamics observed in neural data [13]. The GLM parameters are also relatively amenable to interpretation: $\beta_i^{\text{self}}(\tau)$ (or, respectively, $\beta_i^{\text{stim}}(\tau + 1)$) determines how a spike (the stimulus) that happened $\tau$ time bins in the past affects the probability that the neuron will spike in the present. Very negative values of $\beta_i^{\text{self}}(\tau)\mathbf{y}_i(t - \tau)$ or $\beta_i^{\text{stim}}(\tau + 1)\tilde{x}_i(t - \tau d^{\text{stim}})$ suppress spiking at time $t$; very positive values promote spiking.

**2.2.2 Cluster model.** After fitting the single-cell model (3), we cluster the fitted self-interaction filters, $\{\hat{\boldsymbol{\beta}}_1^{\text{self}}, ..., \hat{\boldsymbol{\beta}}_N^{\text{self}}\}$, to produce cell-type assignments for each neuron. See Section B.1.1 in S1 Text for consideration of the case where all of $\{\hat{\boldsymbol{\beta}}_1, ..., \hat{\boldsymbol{\beta}}_N\}$, including the stimulus filters, are clustered.

To cluster the coefficient estimates, we fit a Gaussian mixture model (GMM) with weights $\pi_k$, means $\boldsymbol{\mu}_k^{\text{self}}$, and covariances $\Sigma_k^{\text{self}}$, $k \in \{1, ..., K\}$. The weights must sum to 1 ($\sum_{k=1}^{K} \pi_k = 1$), and additionally we assume that the covariance matrices are diagonal. We fit the GMM with the `GaussianMixture` class from `scikit-learn` [11], enforcing diagonal covariance matrices (`covariance_type='diag'`), initializing the optimizer with the solution of K-means clustering (`init_params='kmeans'`), and choosing the best optimization of those obtained from 20 random initializations (`n_init = 20`). Collectively, we refer to these

parameters of the clustering as $\Omega_K \equiv \{\pi_1, ... \pi_K, \boldsymbol{\mu}_1^{\text{self}}, ..., \boldsymbol{\mu}_K^{\text{self}}, \Sigma_1^{\text{self}}, ..., \Sigma_K^{\text{self}}\}$. Thus we write the clustering as:

$$\hat{\Omega}_K \leftarrow \underset{\Omega_K}{\arg\max} \sum_{i=1}^{N} \log \sum_{k=1}^{K} \pi_k f(\hat{\boldsymbol{\beta}}_i^{\text{self}}; \boldsymbol{\mu}_k^{\text{self}}, \Sigma_k^{\text{self}}). \tag{5}$$

Once we have fit the GMM to the parameter estimates $\{\hat{\boldsymbol{\beta}}_1^{\text{self}}, ..., \hat{\boldsymbol{\beta}}_N^{\text{self}}\}$, the maximum likelihood estimate (MLE) for the cell type of each neuron is

$$\hat{k}_i \leftarrow \underset{k}{\arg\max} \, \hat{\pi}_k f(\hat{\boldsymbol{\beta}}_i^{\text{self}}; \hat{\boldsymbol{\mu}}_k^{\text{self}}, \hat{\Sigma}_k^{\text{self}}), \; i = 1, ..., N. \tag{6}$$

**Algorithm 1**: Sequential method.

```
Data: x_i(t), y_i(t), t = 1, ..., T_i, i = 1, ..., N, K, d^stim
Result: Ω̂_K ≡ {π̂_k, μ̂_k^self, Σ̂_k^self}, k = 1,...,K, β̂_i, k̂_i, i = 1,...N
/* Step 1: Estimate response model parameters β_1, ..., β_N*/
for a logarithmically spaced grid of λ^stim, λ^self do
  VALL(λ^stim, λ^self) ← 0
  for i = 1, ..., N do
    Partition the ith neuron's data into L equally-sized bins of
    adjacent time points, (x_i^1,...,x_i^L) and (y_i^1,...,y_i^L)
    for l = 1,..,L do
      β̂_i(λ^stim, λ^self) ← arg max_β { L/((L-1)T_i) Σ_{l'≠l} log P_SC(y_i^l'|x_i^l';β) − ½λ^stim||β^stim||_2^2 − ½λ^self||β^self||_2^2 }.
      VALL(λ^stim, λ^self) ← VALL(λ^stim, λ^self) + 1/(NT_i) log P_SC(y_i^l|x_i^l;β̂_i(λ^stim, λ^self)).
    end
  end
end
λ̂^stim, λ̂^self ← arg max_{λ^stim,λ^self} VALL(λ^stim,λ^self).
β̂_i ← arg max_β log P_SC(y_i|x_i;β) + ½λ̂^stim||β^stim||_2^2 + ½λ̂^self||β^self||_2^2, i = 1, ..., N.
/* Step 2: Cluster estimated β_i^self into cell types k_i.        */
Ω̂_K ← arg max_{Ω_K} Π_{i=1}^N Σ_{k=1}^K π_k f(β̂_i^self;μ_k^self, Σ_k^self) (Fit a standard GMM)
k̂_i ← arg max_k π̂_k f(β̂_i^self;μ̂_k^self, Σ̂_k^self), i = 1,...,N
```

## 2.3 Simultaneous method

In the simultaneous method, we define a generative model for the data $x_i(t)$, $y_i(t)$ over all neurons, given that there are $K$ classes. In this model, the response to stimuli will be determined by latent variables, which we denote by $\boldsymbol{\beta}_i = [\boldsymbol{\beta}_i^{\text{stim}}, \boldsymbol{\beta}_i^{\text{self}}, \beta_i^0]^\top$, that are distributed according to a Gaussian distribution, $f(\boldsymbol{\beta}; \boldsymbol{\mu}_k, \Sigma_k)$, given membership in class $k$. Class membership is given by the categorical distribution with parameters $\pi_1, \ldots, \pi_K$.

We take $\{\mathbf{x}_1, \ldots, \mathbf{x}_N\}$ as fixed inputs, and let $\Omega_K = \{\pi_1, ..., \pi_K, \boldsymbol{\mu}_1^{\text{self}}, ..., \boldsymbol{\mu}_K^{\text{self}}, \Sigma_1^{\text{self}}, ..., \Sigma_K^{\text{self}}\}$ denote the set of all parameters of the generative model with $K$ classes. We use the same symbols for the simultaneous and sequential methods to emphasize their shared structure and facilitate comparisons.

We write the joint likelihood of a combination of latent variables and data for the $i$th neuron as

$$P_{\text{joint}}(k, \boldsymbol{\beta}, \mathbf{y}_i|\mathbf{x}_i; \Omega_K) = P(\mathbf{y}_i|k, \boldsymbol{\beta}, \mathbf{x}_i; \Omega_K)P(\boldsymbol{\beta}|k, \mathbf{x}_i; \Omega_K)P(k|\mathbf{x}_i; \Omega_K). \tag{7}$$

We then make the following assumptions:

1. $P(\mathbf{y}_i|k, \boldsymbol{\beta}, \mathbf{x}_i; \Omega_K) \equiv P_{SC}(\mathbf{y}_i|\mathbf{x}_i; \boldsymbol{\beta})$, where $P_{SC}$ was defined in (1) and (2). Thus, the spiking response of the $i$th neuron given its stimulus and single-cell parameters $\boldsymbol{\beta}_i$ is conditionally independent of $k_i$ and is not a function of $\Omega_K$.

2. $P(\boldsymbol{\beta}|k, \mathbf{x}_i; \Omega_K) \equiv f(\boldsymbol{\beta}; \boldsymbol{\mu}_k, \Sigma_k)$, so that the single-cell parameters for the $i$th neuron are independent of $\mathbf{x}_i$.

3. The matrix $\Sigma_k$ is diagonal for $k = 1, \ldots, K$, so that the elements of $\boldsymbol{\beta}_i$ are mutually independent, given $k_i$.

4. $P(k|\mathbf{x}_i; \Omega_K) = P(k; \Omega_K) \equiv \pi_k$. Thus the cluster label for the $i$th neuron is independent of $\mathbf{x}_i$, $\boldsymbol{\mu}_k$, and $\Sigma_k$.

With these assumptions, we re-write (7) as

$$P_{\text{joint}}(k, \boldsymbol{\beta}, \mathbf{y}_i|\mathbf{x}_i; \Omega_K) = P_{SC}(\mathbf{y}_i|\mathbf{x}_i; \boldsymbol{\beta})f(\boldsymbol{\beta}; \boldsymbol{\mu}_k, \Sigma_k)\pi_k. \tag{8}$$

Recalling that we used superscripts to represent the components of $\boldsymbol{\beta}_i \equiv [\boldsymbol{\beta}_i^{\text{stim}}, \boldsymbol{\beta}_i^{\text{self}}, \beta_i^0]^\top$, we denote the components of the cluster means as $\boldsymbol{\mu}_k = [\boldsymbol{\mu}_k^{\text{stim}}, \boldsymbol{\mu}_k^{\text{self}}, \mu_k^0]^\top$, and likewise for the covariances, with $\Sigma_k^{\text{stim}}, \Sigma_k^{\text{self}}, (\sigma_k^0)^2$ diagonally stacked to form $\Sigma_k$ (recall that $\Sigma_k$ is diagonal).

As with the sequential method, we assume that $\boldsymbol{\beta}_i^{\text{stim}}$ and $\beta_i^0$ are independent of the cell type $k_i$. (See Section B.1.2 in S1 Text for the case where all parameters are cell-type-dependent.) Specifically:

- $\boldsymbol{\mu}_1^{\text{stim}} = \ldots = \boldsymbol{\mu}_K^{\text{stim}} = 0, \Sigma_1^{\text{stim}} = \ldots = \Sigma_K^{\text{stim}} = (1/\lambda^{\text{stim}}) * I$. This amounts to applying $\ell_2$ regularization to the stimulus filters, as in the sequential method.

- The prior for $\beta_i^0, f(\beta_i^0; \mu_k^0, (\sigma_k^0)^2)$, is flat. That is, no regularization is applied to the offset term; we can think of this as taking $(\sigma_k^0)^2 \to \infty$.

Thus (8) may be further factorized as

$$P_{\text{joint}}(k, \boldsymbol{\beta}, \mathbf{y}_i|\mathbf{x}_i; \Omega_K) \propto P_{SC}(\mathbf{y}_i|\mathbf{x}_i; \boldsymbol{\beta})f(\boldsymbol{\beta}_i^{\text{stim}}; 0, (1/\lambda^{\text{stim}}) * I)f(\boldsymbol{\beta}_i^{\text{self}}; \boldsymbol{\mu}_k^{\text{self}}, \Sigma_k^{\text{self}})\pi_k. \tag{9}$$

This hierarchical model has two *hyperparameters*, $K$ and $\lambda^{\text{stim}}$, a set of *global parameters* describing the cell types, $\Omega_K$, and *latent variables* for the $i$th neuron, $k_i$ and $\boldsymbol{\beta}_i$. Note that there is no $\lambda^{\text{self}}$ hyperparameter for $\ell_2$ regularization of $\boldsymbol{\beta}_i^{\text{self}}$ with a Gaussian prior, as the the clustering induces regularization on these parameters.

To obtain the likelihood of the data, we marginalize over the latent variables:

$$P(\mathbf{y}_i|\mathbf{x}_i; \Omega_K) = \sum_{k=1}^{K} \int P_{\text{joint}}(k, \boldsymbol{\beta}, \mathbf{y}_i|\mathbf{x}_i; \Omega_K)d\boldsymbol{\beta}. \tag{10}$$

We further assume that the spiking activity of each neuron is independent, and so the likelihood of an entire dataset is simply a product over neurons:

$$P(\mathbf{y}_1, \ldots, \mathbf{y}_N|\mathbf{x}_1, \ldots, \mathbf{x}_N; \Omega_K) = \prod_{i=1}^{N} P(\mathbf{y}_i|\mathbf{x}_i; \Omega_K). \tag{11}$$

**2.3.1 EM algorithm.** We adapt the expectation-maximization (EM) algorithm [8] for the generative model in (9) to find the MLE of $\Omega_K$,

$$\Omega_K^* = \arg\max_{\Omega_K} P(\mathbf{y}_1, ..., \mathbf{y}_N | \mathbf{x}_1, ..., \mathbf{x}_N; \Omega_K) \tag{12}$$

(see (11)). Each $\mathbf{y}_i$ and $\mathbf{x}_i$ correspond to a single independent neuron (each $\mathbf{y}_i$ is a sample, in common EM language), with associated latent variables $k_i$ and $\boldsymbol{\beta}_i$; the global parameters of our model are $\Omega_K$.

We outline the two steps of the EM algorithm here, and unpack them in detail in Section A in S1 Text:

- E-step: for $i = 1, \ldots, N$, we compute the posterior distribution over latent variables $k_i, \boldsymbol{\beta}_i$, given the data and the current estimate of global parameters $\hat{\Omega}_K$. We call this $Q_i(k, \boldsymbol{\beta})$:

$$
\begin{aligned}
Q_i(k, \boldsymbol{\beta}) &\equiv P(k, \boldsymbol{\beta} | \mathbf{x}_i, \mathbf{y}_i; \hat{\Omega}_K) \\
&= \frac{P_{\text{joint}}(k, \boldsymbol{\beta}, \mathbf{y}_i | \mathbf{x}_i; \hat{\Omega}_K)}{P(\mathbf{y}_i | \mathbf{x}_i; \hat{\Omega}_K)} = \frac{P_{\text{joint}}(k, \boldsymbol{\beta}, \mathbf{y}_i | \mathbf{x}_i; \hat{\Omega}_K)}{\sum_{k=1}^{K} \int P_{\text{joint}}(k, \boldsymbol{\beta}, \mathbf{y}_i | \mathbf{x}_i; \hat{\Omega}_K) d\beta}.
\end{aligned}
\tag{13}
$$

Due to the complexity of the denominator of the right hand side of (13), we cannot compute the right-hand side exactly. Therefore we employ a weighted Gaussian approximation for $P_{\text{joint}}(k, \boldsymbol{\beta}, \mathbf{y}_i | \mathbf{x}_i; \hat{\Omega}_K)$:

$$P_{\text{joint}}(k, \boldsymbol{\beta}, \mathbf{y}_i | \mathbf{x}_i; \hat{\Omega}_K) \approx Z_{i,k} f(\boldsymbol{\beta}; \boldsymbol{m}_{i,k}, c_{i,k}). \tag{14}$$

This approximation allows us to simplify (13) as follows:

$$
\begin{aligned}
&\frac{P_{\text{joint}}(k, \boldsymbol{\beta}, \mathbf{y}_i | \mathbf{x}_i; \hat{\Omega}_K)}{\sum_{k=1}^{K} \int P_{\text{joint}}(k, \boldsymbol{\beta}, \mathbf{y}_i | \mathbf{x}_i; \hat{\Omega}_K) d\boldsymbol{\beta}} \\
&\qquad\qquad \approx \frac{Z_{i,k} f(\boldsymbol{\beta}; \boldsymbol{m}_{i,k}, c_{i,k})}{\sum_{k=1}^{K} \int Z_{i,k} f(\boldsymbol{\beta}; \boldsymbol{m}_{i,k}, c_{i,k}) d\boldsymbol{\beta}} = \frac{Z_{i,k}}{\sum_{k=1}^{K} Z_{i,k}} f(\boldsymbol{\beta}; \boldsymbol{m}_{i,k}, c_{i,k}).
\end{aligned}
\tag{15}
$$

The normalized weights $\frac{Z_{i,k}}{\sum_{k=1}^{K} Z_{i,k}}$ appear repeatedly in what follows, so we denote them

$$\tilde{Z}_{i,k} \equiv \frac{Z_{i,k}}{\sum_{k=1}^{K} Z_{i,k}}. \tag{16}$$

This, along with (13) and (15), allows us to write

$$Q_i(k, \boldsymbol{\beta}) \approx \tilde{Z}_{i,k} f(\boldsymbol{\beta}; \boldsymbol{m}_{i,k}, c_{i,k}). \tag{17}$$

Our E-step now consists of finding the optimal $Z_{i,k}, \boldsymbol{m}_{i,k}, c_{i,k}$ to make the approximation in (17) as good as possible (more detail in Section A.1 in S1 Text). We note that many other approximations may be suitable here, such as mean field variational inference [14], but have chosen this one for its simplicity and the low computational cost of the resulting algorithm (see Section A.3 in S1 Text for details).

- M-step: update the estimate of the global parameters $\hat{\Omega}_K$ by maximizing an approximation to a lower bound on $\log(P(\mathbf{y}_1, ..., \mathbf{y}_N | \mathbf{x}_1, ..., \mathbf{x}_N, \hat{\Omega}_K))$:

$$\hat{\Omega}_K = \arg\max_{\Omega_K} \sum_{i=1}^{N} \sum_{k=1}^{K} \int \tilde{Z}_{i,k} f(\boldsymbol{\beta}; \boldsymbol{m}_{i,k}, c_{i,k}) \log P_{\text{joint}}(k, \boldsymbol{\beta}, \mathbf{y}_i | \mathbf{x}_i; \Omega_K) d\boldsymbol{\beta}. \quad (18)$$

**Latent variable estimates.** Once the EM algorithm has converged to a final point estimate of $\Omega_K$, which we denote $\Omega_K^*$, we may also want to compute point estimates of $k_i$ and $\boldsymbol{\beta}_i$. While it is tempting to estimate both simultaneously as $(\hat{k}_i, \hat{\boldsymbol{\beta}}_i) = \arg\max_{k, \boldsymbol{\beta}} P_{\text{joint}}(k, \boldsymbol{\beta}, \mathbf{y}_i | \mathbf{x}_i; \Omega_K^*)$, this approach will tend to assign neurons to clusters with smaller estimated variances.

Instead, we estimate $k_i$ for each neuron by maximizing the likelihood with $\boldsymbol{\beta}_i$ marginalized out:

$$
\begin{aligned}
\hat{k}_i &= \arg\max_k \int P_{\text{joint}}(k, \boldsymbol{\beta}, \mathbf{y}_i | \mathbf{x}_i; \Omega_K^*) d\boldsymbol{\beta} \approx \arg\max_k \int Z_{i,k} f(\boldsymbol{\beta}; \boldsymbol{m}_{i,k}, c_{i,k}) d\boldsymbol{\beta} \\
&= \arg\max_k Z_{i,k}.
\end{aligned}
\quad (19)
$$

After estimating $k_i$, we estimate $\hat{\boldsymbol{\beta}}_i$ by maximizing the likelihood with respect to it,

$$
\begin{aligned}
\hat{\boldsymbol{\beta}}_i &= \arg\max_{\boldsymbol{\beta}} P_{\text{joint}}(\hat{k}_i, \boldsymbol{\beta}, \mathbf{y}_i | \mathbf{x}_i; \Omega_K^*) \approx \arg\max_{\boldsymbol{\beta}} Z_{i,\hat{k}_i} f(\boldsymbol{\beta}; \boldsymbol{m}_{i,\hat{k}_i}, c_{i,\hat{k}_i}) \\
&= \boldsymbol{m}_{i,\hat{k}_i}.
\end{aligned}
\quad (20)
$$

**Summary of EM algorithm.** The overall EM algorithm is spelled out in Algorithm 2. We perform this algorithm 20 times from different random initializations and use the results from the optimization with the lowest loss (see Section 2.4). This choice mirrors that used for the sequential method (see Section 2.2.2). We also note that the initialization of the cluster parameters ($\Omega_K \leftarrow$ GMM fit to $\hat{\boldsymbol{\beta}}_1, ..., \hat{\boldsymbol{\beta}}_N$) uses the same settings as for the sequential method, except that only 10 random initializations are used.

**Algorithm 2**: EM algorithm for simultaneous model. See Sections A.1 and A.2 in S1 Text for detailed derivations of each step.

```
Data: x_i(t), y_i(t) for t = 1, ..., T_i, i = 1, ..., N, K, d^stim
Result: π̂_k, μ̂_k, Σ̂_k for k = 1,...,K, k̂_i, β̂_i for i = 1,...,N
For i ∈ {1, ..., N}, β̂_i ← arg max_β P(y_i|x_i, β), using (2)
Ω_K ← GMM fit to β̂_1,...,β̂_N
repeat
  /* E-Step: approximate distributions over latent variables k_i, β_i */
  for i = 1, ..., N do
    m_{i,k} = arg max_β log P_joint(k, β, y_i|x_i; Ω_K)  k = 1, ..., K
    c_{i,k}^{-1} = -diag(∇²_β log P_joint(k,β,y_i|x_i;Ω_K))|_{β=m_{i,k}}  k = 1,...,K
    Z_{i,k} ← P_joint(k,β,y_i|x_i;Ω_K)|_{β=m_{i,k}} √((2π)^{dim(β)}|c_{i,k}|)  k = 1,...,K
    Z̃_{i,k} ← Z_{i,k}/∑_{k'=1}^{K} Z_{i,k'}  k = 1,...,K
  end
  /* M-Step: update estimates of global parameters π̂_k, μ̂_k, Σ̂_k */
  for k = 1, ..., K do
    π̂_k ← (1/N)∑_{i=1}^{N} Z̃_{i,k}
    μ̂_k^{self} ← (∑_{i=1}^{N} Z̃_{i,k} m_{i,k}^{self}) / (∑_{i=1}^{N} Z̃_{i,k})
```

$$\hat{\Sigma}_k^{\text{self}} \leftarrow \frac{\sum_{i=1}^N \tilde{Z}_{i,k} \text{diag}(c_{i,k}^{\text{self}} + \boldsymbol{m}_{i,k}^{\text{self}} \boldsymbol{m}_{i,k}^{\text{self}\top} - \hat{\boldsymbol{\mu}}_k^{\text{self}} \hat{\boldsymbol{\mu}}_k^{\text{self}\top})}{\sum_{i=1}^N \tilde{Z}_{i,k}}$$

**end**
**until** *convergence*;
$\hat{k}_i \leftarrow \arg\max_k \tilde{Z}_{i,k}, \ i = 1, ..., N$
$\hat{\boldsymbol{\beta}}_i \leftarrow \boldsymbol{m}_{i,\hat{k}_i}, \ i = 1, ..., N$

## 2.4 Model selection

Since the true value of $K$ is generally unknown, we consider two separate criteria for estimating $\hat{K}$, as well as the other hyperparameter $\hat{\lambda}^{\text{stim}}$ for the simultaneous method: Bayesian information criterion (BIC) and cross-validation log-likelihood on held-out neurons (CVLL, considered in Section E.1 in S1 Text). For the simultaneous method, both criteria are applied to the approximate marginal log-likelihood for the hierarchical model, $\text{LL}_i \equiv \log \sum_{k=1}^K Z_{i,k} \approx \log P(\boldsymbol{y}_i | \boldsymbol{x}_i; \hat{\Omega}_k)$, as computed in (C) in S1 Text. For the sequential method, both criteria are applied to the GMM log-likelihood, $\text{LL}_i \equiv \log \sum_{k=1}^K \hat{\pi}_k f(\hat{\boldsymbol{\beta}}_i; \hat{\boldsymbol{\mu}}_k, \hat{\Sigma}_k)$. (The $\lambda$ hyperparameters were already selected using (4).) Both BIC and CVLL are evaluated for a range of $K$ (and $\lambda^{\text{stim}}$ for the simultaneous method), to select the optimal set of hyperparameters.

We use the following heuristic for BIC:

$$BIC = \sum_{i=1}^N \text{LL}_i - \frac{\text{dof}(\hat{\Omega}_K)}{2} \log(N). \tag{21}$$

Above, $\text{dof}(\hat{\Omega}_K)$ is the number of degrees of freedom in $\hat{\Omega}_K$, equal to $\text{dof}(\{\hat{\pi}_k\}) + \text{dof}(\{\hat{\Sigma}_k\}) + \text{dof}(\{\hat{\boldsymbol{\mu}}_k\}) = K - 1 + K * T^{\text{self}} + K * T^{\text{self}} = K * (2 * T^{\text{self}} + 1) - 1$ (recall that $\sum_{k=1}^K \hat{\pi}_k = 1$ and that each $\hat{\Sigma}_k$ is diagonal).

We note here that these model selection criteria assume a specific form of model, and thus do not admit direct comparisons between the simultaneous and sequential methods, or with the alternative model we consider in Section B in S1 Text. All plots displaying model criteria are normalized so that the best value attained is at 0 to facilitate judging the scale of the plot. All direct comparisons between methods must therefore rely on the evaluation metrics we consider in the following Section.

## 2.5 Evaluation metrics

Throughout the results, we use several metrics to assess how accurately each method estimates clustering and parameters.

To assess clustering, we use the adjusted Rand score (ARS) (see [15] 25.1.2.2), which compares two labelings of a set of neurons. The ARS is symmetric in its inputs, equals 1 if the two labelings agree perfectly, and equals 0 if the two labelings are as similar as expected by chance. In Section 3.1, we compute ARS to compare estimated clusters to ground truth labels of simulated data. In Section 3.2, we use ARS to compare two sets of estimated clusters, obtained using two different sets of training neurons. This gives us a measure of how robust the clustering is to the inclusion or exclusion of subsets of neurons from training.

To assess accuracy of parameter estimation, we use the root-mean-squared error (RMS), $\sqrt{\frac{1}{T^{\text{stim}}} \sum_{t=1}^{T^{\text{stim}}} \left(\beta^{\text{stim}}(t) - \hat{\beta}^{\text{stim}}(t)\right)^2}$, when ground truth is available. For the self-interaction filter, we exclude coefficients whose true value is less than $-4$, as parameter inaccuracies in this regime have a minimal effect on the likelihood.

When the ground truth is not available, we use two separate metrics to evaluate GLM parameter estimates for a given neuron based on how well they explain held-out test data. We use the average negative log-likelihood (ANLL) to evaluate the performance of each GLM on test data, in terms of the same quantity that was used to train it,

$$ANLL_i \equiv -\frac{1}{T_i^{\text{test}}} \log(P_{SC}(\mathbf{y}_i^{\text{test}}|\mathbf{x}_i^{\text{test}}; \hat{\boldsymbol{\beta}}_i)). \tag{22}$$

We also compute the explained variance ratio ($EV_{ratio}$), following [6]. To compute $EV_{ratio}$, we require that the test data consists of neural responses to repeated presentations of a single stimulus waveform. Calculating the $EV_{ratio}$ for the parameter estimate of a given neuron then consists of three steps:

1. *Simulate the response of a GLM with the estimated parameters to the test stimulus.* Because our model is stochastic, we simulate a large number (3000) of responses by repeatedly sampling from (2), and average the simulated spike trains to produce the *average model prediction.*

2. *Smooth the spiking response to each presentation of the test stimulus, as well as the average model prediction, with a Gaussian kernel with a standard deviation of 10ms.* We refer to the neuron's smoothed response to the *j*th of *P* presentations of the test stimulus as *stPSTH_j*, which is an abbreviation for *single peristimulus time histogram*. We average these quantities to obtain the smoothed average response, defined as $PSTH_D \equiv \frac{1}{P}\sum_{j=1}^{P} stPSTH_j$. We let $PSTH_M$ denote the smoothed average model prediction.

3. *Define $EV_{ratio}$ as the ratio of the trial averaged explained variance between $PSTH_M$ and $stPSTH_j$ to that between $PSTH_D$ and $stPSTH_j$.* That is,

$$EV_{ratio} = \frac{\sum_{j=1}^{P} EV(stPSTH_j, PSTH_M)}{\sum_{j=1}^{P} EV(stPSTH_j, PSTH_D)}, \tag{23}$$

where

$$EV(PSTH_1, PSTH_2) = \frac{var(PSTH_1) + var(PSTH_2) - var(PSTH_1 - PSTH_2)}{var(PSTH_1) + var(PSTH_2)}. \tag{24}$$

Note that (24) will equal 1 if $PSTH_1 = PSTH_2$, and will equal zero if $PSTH_1$ and $PSTH_2$ are independent; it can be seen as the scaled covariance across time points between the $PSTH_1$ and $PSTH_2$. A very high value of $EV_{ratio}$ (near 1) would thus indicate that the average model prediction covaries across time with the individual trial responses almost as well as the trial-averaged response (and thus that the model fits the data well); a very low value (near 0) would indicate that the average model prediction has low covariance with the individual trial responses.

## 3 Results

In Section 2, we detailed two methods to identify cell types from neural spiking responses, the sequential approach and our simultaneous approach. The sequential approach consists of individually tuning the parameters of a generalized linear model (GLM, Section 2.2.1) to fit each neuron's responses and then clustering those parameters (Section 2.2). The simultaneous approach makes use of a hierarchical probabilistic framework to simultaneously estimate both the GLM parameters and their cluster labels (Section 2.3). Crucially, the simultaneous

approach "borrows strength" from other neurons' data, allowing for improved estimates of the GLM parameters, in addition to improved estimates of cell types.

In this section, we first demonstrate that the simultaneous method recovers the ground truth parameters used to generate simulated data (Section 3.1). We then apply it to the Allen Cell Types Database collected by the Allen Institute for Brain Science [9] (Section 3.2). Specifically, we model the spiking response of chemically-isolated neurons in mouse primary visual cortex to an injected pink noise current. This dataset provides an excellent benchmark for these methods, as it contains large amounts of high-quality data collected from many different neurons.

Importantly, these neurons were chosen from a variety of different transgenic lines in an attempt to sample a diverse set of cells that will be useful for characterizing cell types. To this end, the transgenic line of each recorded neuron has been made available in the dataset, as well as information about the cell's location and morphology. This enables us to compare the putative cell types we extract to these other recorded properties of neurons, (Section 3.2.2, see [6] for a similar analysis on the same data).

We demonstrate that our novel simultaneous method produces single-cell models that generally predict spiking responses in the Allen Cell Types Database better than individually fitted models. We further show that the choice of metric used to quantify this improvement leads to different implications regarding which neurons' models are most improved, and how the improvement scales with the number of neurons and amount of data used per neuron. We demonstrate that the clusters of Allen Cell Types Database neurons discovered by our simultaneous method have properties that generally make them amenable to interpretation. We also interpret the discovered clusters by comparing membership in each with the available information about each cell's transgenic line, location, and morphology. For all of these analyses, we use the sequential method and its individually fit GLMs for comparison.

Throughout, we fix the dimensions of $\boldsymbol{\beta}_i^{\text{stim}}$ and $\boldsymbol{\beta}_i^{\text{self}}$ to $T^{\text{stim}} = 10$ and $T^{\text{self}} = 20$, respectively, and use a downsampling factor of $d^{stim} = 5$ for stimulus filters. We use 2 ms time bins, so this corresponds to filtering the last 40 ms of spiking history and the last 100 ms of stimulus.

## 3.1 Application to simulated datasets shows that the simultaneous method accurately recovers ground truth parameters

First, we compare how well each method recovers the true parameters of a generative model from simulated data. To create these data, we use the same stimuli that were presented in the Allen Cell Types Database, and simulate responses of GLMs with identical, fixed stimulus filters $\boldsymbol{\beta}_i^{\text{stim}}$ and offsets $\beta_i^0 = -5$, and with self-interaction filters $\boldsymbol{\beta}_i^{\text{self}}$ sampled from a GMM with equal cluster sizes $\pi_1 = \cdots = \pi_K$ (for simplicity we just simulate 40 neurons per cluster) and fixed $\boldsymbol{\mu}_k^{\text{self}}$ (see Fig 2A and 2B for fixed parameters, Section D in S1 Text for more details). We vary the number of clusters $K$ and the within-cluster variance $\sigma^2$ ($\sigma^2 I = \Sigma_k^{\text{self}}, \ \forall k \in \{1, ..., K\}$).

We fit the simultaneous and sequential methods to the data, and then plot several accuracy metrics for $\sigma \in [10^{-2}, 10^{-5/6}]$. We use an "oracle" approach to determine the hyperparameters for both methods, using the $K$ that was used to generate the data, and $\lambda$ hyperparameters that most accurately recover $\boldsymbol{\beta}$ (by RMS, see Section 2.5) on 10 held-out "oracle" datasets. Because only the sequential method applies $\ell_2$ regularization to self-interaction filters, those estimates particularly are biased towards 0 (for example, see Fig 2B), especially for very negative filter coefficients. However, the exact magnitude of such negative coefficients does not affect the GLM's spiking likelihood much, so we exclude these coefficient estimates from measures of self-interaction filter accuracy (see Section 2.5 for details).

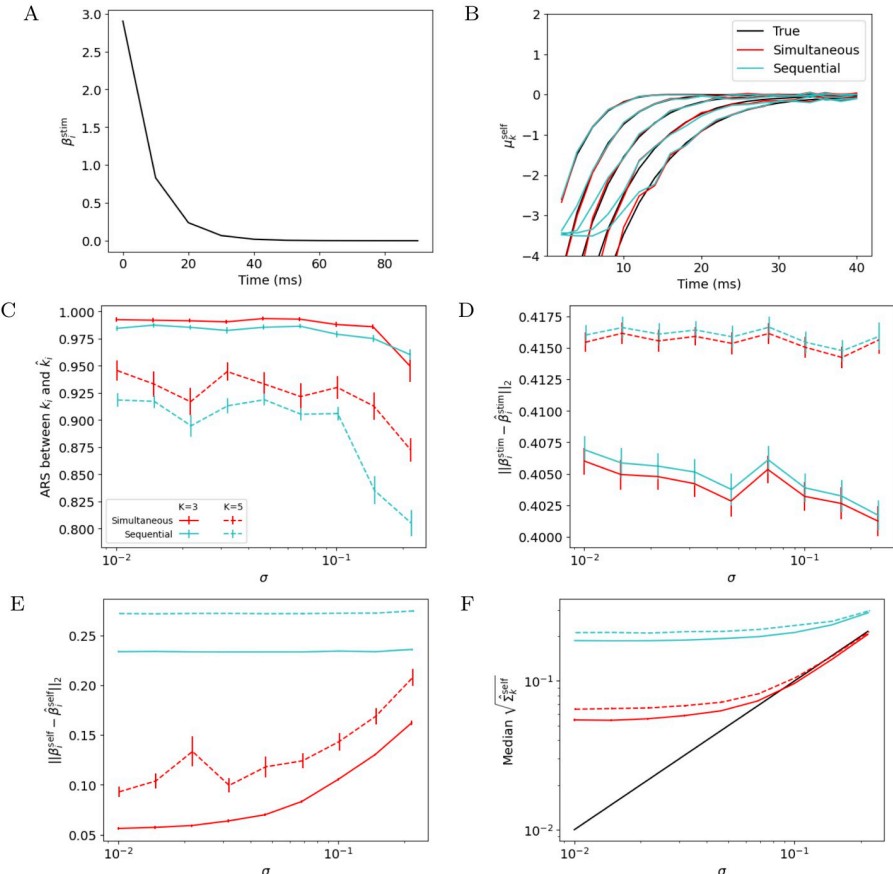

**Fig 2. Performance of sequential and simultaneous methods on simulated data.** A: The true stimulus filter $\boldsymbol{\beta}_i^{\text{stim}}$ for each simulated neuron. B: The true cluster means $\boldsymbol{\mu}_k^{\text{self}}$ used to generate simulated datasets, and those estimated by the sequential and simultaneous methods, $\hat{\boldsymbol{\mu}}_k^{\text{self}}$, fit with the correct $K = 5$. C-F: Mean ± SEM over 50 simulated datasets of accuracy measures, as a function of $\sigma$ and shown for both $K = 3$ and $K = 5$. When $K = 3$, the three clusters with leftmost $\boldsymbol{\mu}_k^{\text{self}}$ in panel B are used. For each condition ($K$ and $\sigma$) and for each measure of accuracy, the simultaneous method's performance is statistically significantly better than that of the sequential method, except for ARS with $K = 3$ and $\sigma = 10^{-5/6}$ (evaluated using the Wilcoxon signed rank test: uncorrected P-value < 0.002).

Our simultaneous method outperforms the sequential method in terms of clustering accuracy (Fig 2C), recovery of single-cell parameters $\beta_i^{\text{stim}}$ (Fig 2D) and $\beta_i^{\text{self}}$ (Fig 2E), and estimation of within-cluster variance (Fig 2F). Note that as $\sigma \to 0$, the estimated cluster spread saturates, reflecting the width of the posterior distribution of the clustered GLM parameters $\boldsymbol{\beta}_i^{\text{self}}$. This saturating value is about $\sqrt{40}$ times lower for the simultaneous method because it essentially pools the data across all 40 simulated neurons in each cluster to estimate the posterior for a single one. These results demonstrate that, by borrowing strength across neurons, the simultaneous method provides better estimates of single-cell model parameters and cell types.

To determine how well each method can recover the true $K$, we evaluate the Bayesian information criterion (BIC) for both methods (Fig 3, see Section E.1 in S1 Text for validation loss on held-out neurons). Here we use an oracle to select the $\lambda$ hyperparameters, but fit models with $K = 1, \ldots, 8$ and select the value of $K$ with the highest BIC (see Section 2.4). We see in Fig 3 that the simultaneous method is better able to recover the true number of clusters.

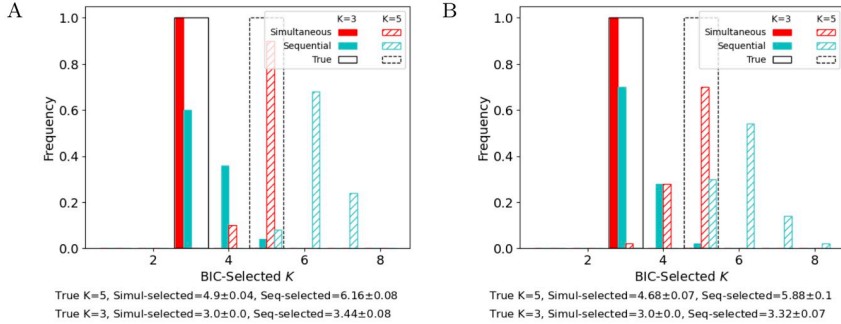

**Fig 3. Model selection of $\hat{K}$.** Frequencies of $\hat{K}$ estimated via Bayesian information criterion over 50 simulated datasets with the same $\mu_k$ as in Fig 2, and $\sigma = 10^{-2}$ (A) or $10^{-5/6}$ (B), the maximum value that does not result in degenerate simulations. Black lines indicate the true value of $K$. The summary below each plot reports the mean ± SEM of the estimated value of $\hat{K}$ across the 50 datasets, for each value of $K$ and each method.

## 3.2 Application to neural data shows that the simultaneous method is a useful tool for modeling and understanding real neural data

To evaluate the simultaneous method on neural data, we apply Algorithm 1 to spiking data recorded from 634 cells in mouse primary visual cortex (Allen Cell Types Database [9], region 'VISp'). These spikes are in response to repeated current injections of pink noise stimuli. Across the entire dataset, only two specific instantiations of pink noise are used, which we refer to as "Noise 1" and "Noise 2"; we selected only cells that received at least three presentations of both, and were labeled with a transgenic cre-line. In order to evaluate how well our method generalizes to new stimuli, all fitting and model selection was done on Noise 1 stimuli, and Noise 2 stimuli were withheld for testing. In applying Algorithm 1, we partitioned each neuron's data into equally-sized bins of adjacent time points so that each element of the partition contains a separate presentation of the Noise 1 stimulus.

We used BIC to perform model selection with the simultaneous method, selecting $\hat{K} = 12$ (Fig 4A, see Section E.1 in S1 Text for model selection using cross-validation on held-out neurons). The discovered cell types are distinct, have smoothed self-interaction filters, and suggest different computational roles for some of the types, as their self-interaction filters have qualitatively different shapes (Fig 4B).

We compare these results to those obtained from individually fitting GLM models for each neuron and then clustering those models' self-interaction filters $\boldsymbol{\beta}_i^{\text{self}}$ (the sequential method). The GLM models are fitted with an $\ell_2$ regularization, with hyperparameters selected using cross-validation, where individual presentations of the stimulus are used to define the partition of time bins (see (4)). Autocorrelations of the residuals (after subtracting the smoothed trial average) indicated very low temporal correlations and the trial correlations of residuals were low for the majority of cells. We find that the estimated $\hat{K}$ is much higher, at least 19 (Fig 4C). The resulting cluster centers for the $K = 12$ (same $K$ as Fig 4B) are all bunched on top of each other near 0 and have higher variances (Fig 4D). We hypothesize that this difference arises from the simultaneous method's ability to borrow strength across different neurons in the same cluster, pulling their parameters closer together toward an improved estimate of their center. By contrast, the sequential method weakly pulls all parameter estimates into a region near 0 with its standard $\ell_2$ penalty, and then fits highly overlapping clusters that are densely packed to fill this region. The cluster weights, $\pi_k$, shown in the legend, also support this

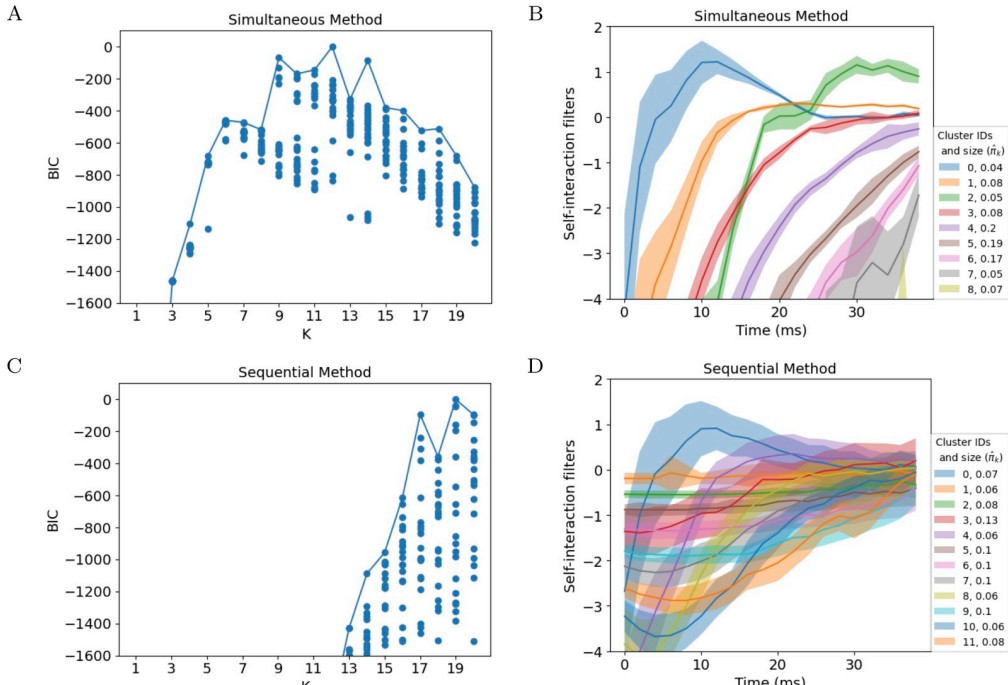

**Fig 4. Allen Cell Types Database: The simultaneous method explains the data with a smaller number of less overlapping clusters.** A: BIC for the simultaneous method over a range of $K$; BIC determines $\hat{K} = 12$ as optimal. Each dot reports the result of running Algorithm 2 from a different random initialization. B: Cluster centers $\boldsymbol{\mu}_k$ of self-interaction filters fit to data using the simultaneous method; shaded region is $\pm\sqrt{diag(\Sigma_k)}$. Only the 9 clusters with at least 20 neurons are shown from the model with BIC-selected $K = 12$. C: By contrast, the standard BIC of a GMM fit to individually fitted self-interaction filters (the sequential method) suggests an optimal $\hat{K}$ of at least 19. Each dot reports the result of performing the GMM fit, (5), from a different random initialization. D: The clusters of self-interaction filters with at least 20 neurons found by the sequential approach with $K = 12$ are bunched up closer to the origin, such that the clusters overlap significantly.

description: the simultaneous method's clusters have much more variable weights ($<0.03$–$0.2$), as well as shapes, compared to the sequential method's (weights 0.06–0.13).

**3.2.1 Generalization performance demonstrates improved parameter estimates.** In order to demonstrate that the differences in parameter estimates between the simultaneous and sequential methods reflect meaningful differences in the descriptions of the underlying biological system, we show that the simultaneous method generalizes better to held-out data. We examine two types of generalization to assess how well each method accomplishes each goal (Section 2.1): namely, generalization to held-out time bins and to held-out neurons. Without knowledge of ground truth single-cell parameters or cell types, this is the best assessment we can make of each method's accuracy. To accomplish this, we partition the neurons into four sets, using each in turn for evaluation after fitting and performing model selection with BIC on the other three, in addition to withholding Noise 2 responses from training.

The simultaneous method discovers individual parameters for each neuron $\hat{\boldsymbol{\beta}}_i$ (20) that allow for better prediction of the held-out responses to Noise 2 than those found by the sequential method (3). We measure this using each fitted GLM's average negative log-likelihood (ANLL) of the held-out data ((22), Fig 5A), as well as its explained variance ratio, ($EV_{ratio}$, how well the mean smoothed model prediction captures the variability in smoothed spike trains across trials, relative to the true cross-trial mean, see (23), Fig 5C). Likewise, we use ARS

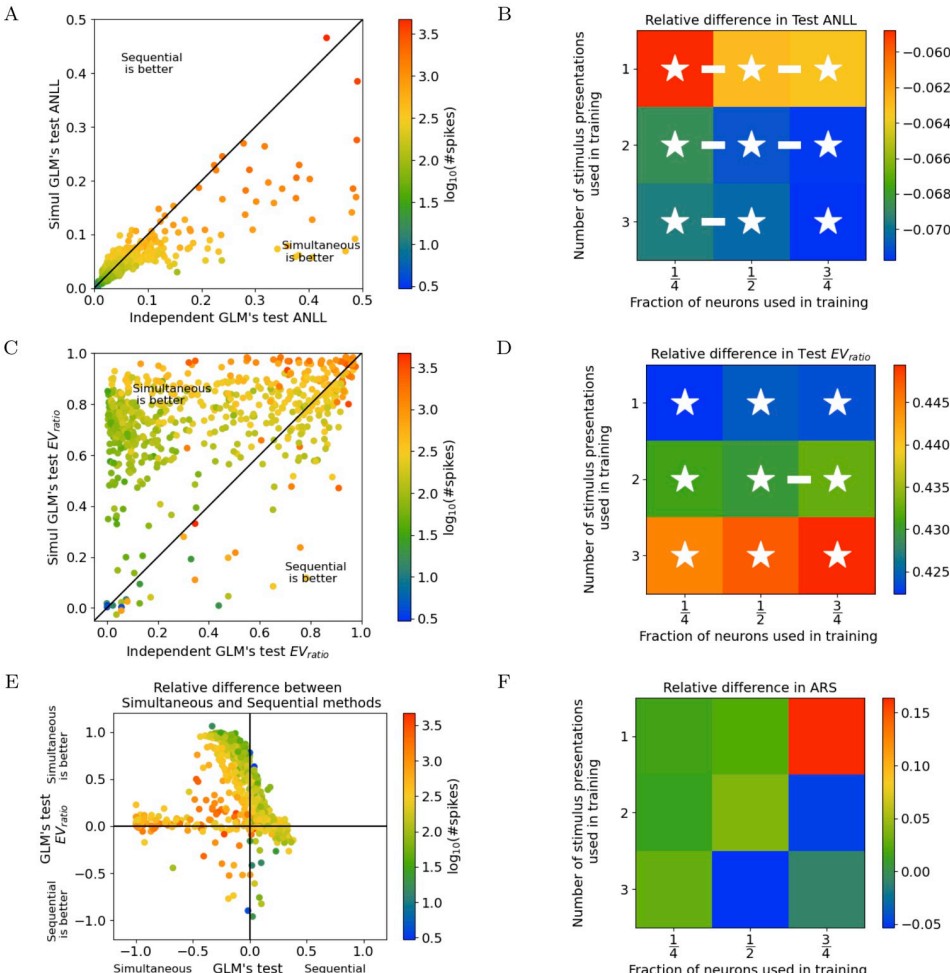

**Fig 5. Allen Cell Types Database generalization performance: The simultaneous method produces single-cell models and clusterings that generalize better, especially when fitted to more neurons.** Additional details about this figure are available in Section C in S1 Text. A: ANLL (lower is better) for each held-out neuron's single-cell model, evaluated on responses to the test stimulus (Noise 2), using the MAP $\hat{\boldsymbol{\beta}}_i$ (20) of the simultaneous method with (hyper) parameters $K$, $\lambda^{\text{stim}}$, $\hat{\Omega}_K$ estimated from the training neurons, versus those found by the sequential method. Color encodes number of spikes for each neuron in the evaluation data. B: Median relative difference between methods in ANLL of held-out neurons, evaluated on responses to the test stimulus, as a function of how many neurons and how much data from each were used in training (more negative values indicate that the simultaneous method is better). White asterisks indicate a significant relative difference; white bars indicate adjacent cases of training data subselection where the relative differences were significantly different. Differences pooled across all vertically (horizontally) adjacent conditions showed a significant, $p = 3 \times 10^{-4}$, ($p = 5 \times 10^{-26}$) trend, with more presentations (neurons) yielding greater improvement by the simultaneous method. C: Same analysis as A, but with $EV_{ratio}$ (see (23); higher values indicate that the simultaneous method is better). D: Same analysis as B, but with $EV_{ratio}$. Pooled vertical differences showed no significant trend ($p > 0.1$); horizontal differences showed a significant ($p = 5 \times 10^{-3}$) trend, with more neurons yielding greater improvement by the simultaneous method. E: Relative differences between methods of $EV_{ratio}$ (shown in C) versus ANLL (shown in A); color encodes number of spikes. Neurons with many (few) spikes show only improved ANLL ($EV_{ratio}$) in the simultaneous method. F: Same analysis as B, but for the similarity of cluster assignments $\hat{k}_i$ between model fits with different held-out neurons, measured by ARS (more positive values indicate that the simultaneous method is better). Pooled vertical differences showed a significant ($p = 5 \times 10^{-2}$) trend, with fewer presentations yielding greater improvement by the simultaneous method; horizontal differences showed no significant trend ($p > 0.1$).

to measure stability of the cluster assignments $\hat{k}_i$ when each of the folds of neurons are held-out from training, and find that the simultaneous method generally finds more stable clusters (positive values in the cells of Fig 5F).

Next, we restricted our analysis to subsets of training trials and subsets of neurons. Recall that each presentation of the Noise 1 stimulus was applied at least three times; we subset the trials so that exactly one, two, or three presentations of Noise 1 was retained per neuron. Furthermore, earlier we split the neurons into four folds and fit the model on three of the four folds; here we instead fit the model on only one, two, or three of the four folds. Finally, we used all Noise 2 presentations for all of the remaining folds as test data.

We find that the relative improvement of the simultaneous method is greater when data from more neurons is used (Fig 5B, 5D, and 5F, although not significantly in Fig 5F). This has a simple interpretation: providing more neurons allows for more borrowing of strength to improve parameter estimates, and thus improves all generalization measures.

The results of varying the number of stimulus presentations per neuron are more ambiguous: ARS benefits the most from the use of the simultaneous method rather than the sequential method when there are fewer presentations, ANLL when there are more, and effects for $EV_{ratio}$ are not significant. These discrepancies can be linked to the observation that different populations of neurons are responsible for the improvement of each metric between methods (Fig 5E). Neurons with fewer spikes in their response will naturally have a good ANLL (see Fig 5A; it is easy to predict zero spikes) and bad $EV_{ratio}$ (see Fig 5C; the inter-spike intervals are very high, so relative jitter in predicted spikes hurts more). Therefore, the simultaneous method improves the $EV_{ratio}$ of these neurons the most, while it improves the ANLL of those with many spikes the most, as in each case those neurons leave the most room for improvement. Given that different populations of neurons are responsible for changes in ANLL and $EV_{ratio}$, it is no surprise that these metrics scale differently with the number of presentations used for training. One potential reason for this is that one of these populations may have a less variable response to the repeated stimulus, yielding a narrower posterior over GLM parameters, than the other. ARS is not computed on a neuron-by-neuron basis, so we cannot easily attribute its difference in scaling to a different population of neurons, but it is clearly assessing performance in a very different way than ANLL and $EV_{ratio}$. Together, these results show that the choice of evaluation metric can drastically affect conclusions about model performance and how it scales with dataset sizes, suggesting that the best practice is to consider many metrics, as we do here.

Additional details about Fig 5 are provided in Section C in S1 Text.

**3.2.2 Comparison to metadata suggests relationships between discovered cell types and measured genetic and anatomical properties of neurons.** The Allen Cell Types Database contains limited morphological, locational, and transcriptomic information about each cell in addition to the electrophysiological recordings. We will now investigate whether the electrophysiological cell types that we discover are related to these metadata, in the same spirit as [2] and [6]. Note, however, that these metadata do not constitute a "ground truth" for functional cell types as we have defined them: they merely provide different dimensions along which neurons can be clustered; any similarities (or lack thereof) between discovered types and the metadata do not suggest better or worse performance of the clustering algorithm. However, as none of the metadata is supplied to either clustering algorithm, any similarities that do exist may provide insights into how functional properties of cells relate to morphological, transcriptomic, or location factors.

Many metadata labels belonged to certain clusters discovered by the simultaneous method much more or less often than expected by chance: namely, dendrite types, most transgenic

lines, and some cortical layers (Fig 6A). Dendrite types and especially cortical layer reflect a quantization of an inherently continuous variable, and accordingly these plots display a horizontal gradient, which is to be expected if each cluster ID has some spread in distribution over the underlying continuous variable. It is worth noting that many columns (attributes) display a vertical gradient (cluster IDs are ordered by the mean values of their estimated self-interaction filters so that $\sum_{t=1}^{T^{\text{self}}} \hat{\mu}_1^{\text{self}}(t) > \cdots > \sum_{t=1}^{T^{\text{self}}} \hat{\mu}_K^{\text{self}}(t)$, and used in Figs 4B, 4D, and 6). This is consistent with the reality that our notion of cell types is also a quantization of an inherently continuous space. Both such quantizations, however, can be useful—for example, certain layers (2/3, 4, and 6b) and transgenic lines (Pvalb, Tlx3, and many others to a lesser extent) are strongly overrepresented in a very small number of clusters and strongly underrepresented in the others. These results suggest that there are meaningful relationships between the functional cell types discovered by our method and these genetic and anatomical factors, beyond what would be expected if cells were uniformly distributed throughout the continuous spaces of functional parameters, dendrite density, depth, and transgenic expression.

Results obtained using the sequential method's clusters are generally similar, albeit reflecting differences in cluster structure seen in Fig 4. Because the cluster means are relatively bunched, the ordering of their indices, $\sum_{t=1}^{T^{\text{self}}} \hat{\mu}_1^{\text{self}}(t) > \cdots > \sum_{t=1}^{T^{\text{self}}} \hat{\mu}_K^{\text{self}}(t)$, is messier than that of the simultaneous method, limiting our ability to observe the vertical gradients described above.

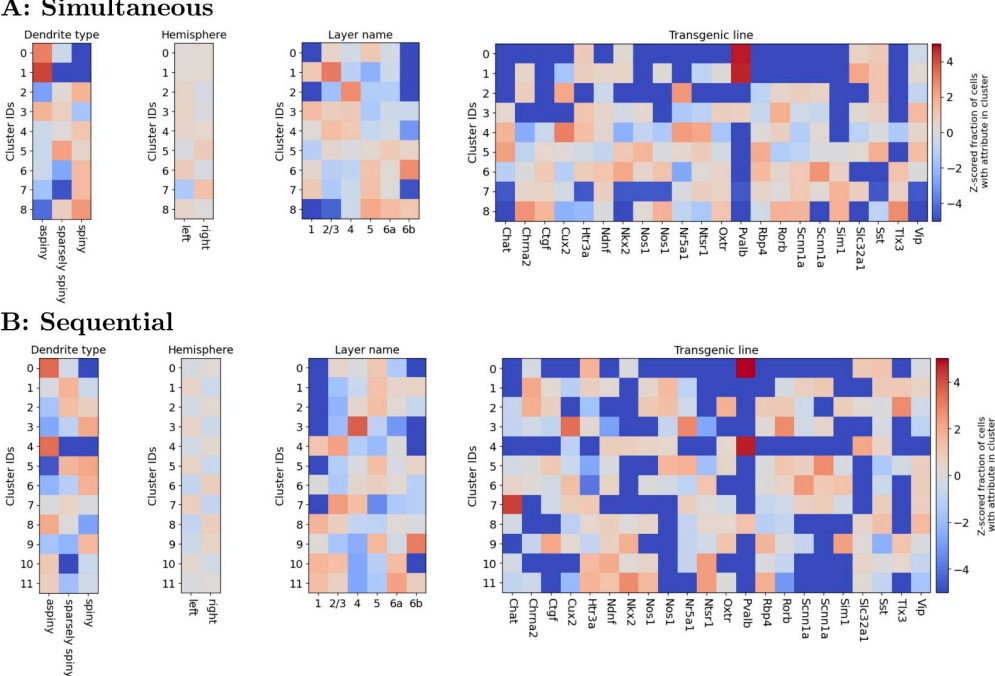

**Fig 6. Allen Cell Types Database metadata is related to discovered cell types.** Z-scored fraction of cells with an attribute in each cluster. Cluster identities in panel A (B) are the same as in Fig 4B (Fig 4D), obtained using the simultaneous (sequential) method (fitted with BIC-selected $K = 12$ clusters, showing only clusters with at least 20 neurons). Attributes are spiny or aspiny dendrites, location (hemisphere and cortical layer), and Cre line. Note the ARS between the cluster and metadata labels in each title. Z-scores are calculated as $Z_i^{(a)} = (\hat{p}_i^{(a)} - \hat{p}_i) / \sqrt{\hat{p}_i^{(a)}(1 - \hat{p}_i^{(a)})/N^{(a)} + \hat{p}_i(1 - \hat{p}_i)/N}$, where $\hat{p}_i$ is the empirical probability that a cell is in cluster $i$ and $\hat{p}_i^{(a)}$ is the empirical probability that a cell with attribute $a$ is in cluster $i$, $N$ is the number of cells, and $N^{(a)}$ is the number of cells with attribute $a$.

We see from Fig 6 that certain attributes tend to be characterized by a small number of clusters in both the simultaneous and sequential methods. For example, neurons in the Pvalb cre line tend to be assigned to clusters 0 and 1 by the simultaneous method, and to clusters 0 and 4 by the sequential method.

## 4 Discussion

In this work, we leverage a hierarchical probabilistic framework to advance our ability to identify cell types from neural responses and improve our models of individual neurons. We find that, even applied to relatively noiseless *in vitro* recordings, our method provides substantial gains over independently fit single-cell models, in terms of its ability to predict the response to a held-out stimulus. We demonstrated that these gains increase as our method is applied to datasets of increasingly many neurons, and highlighted the importance of using multiple evaluation metrics. Compared to clustering individually fitted neuron models, our method discovers cell types that are more robust to the exclusion of different groups of neurons from training, are more amenable to interpretation, and reveal trends of scientific interest in terms of correlations between cluster membership and other information available about each neuron.

We argue that these improvements stem from the simultaneous method's ability to "borrow strength" between the spike trains of different neurons. That is, while both approaches can be thought of as applying regularization to the estimation of single-cell parameters for each neuron, $\hat{\boldsymbol{\beta}}_i$, in the simultaneous case this regularization is informed by the spiking responses of other neurons, bringing more data to bear on the estimation problem.

Compared with using sequential approaches that separately fit single neuron models and then cluster their parameters, using this hierarchical generative model framework requires more advanced statistical methods for parameter estimation and model selection. However, appropriate choices of models and approximations allow for tractable and improved parameter estimation. We make particular choices for the single-cell response model (GLM), which parameters are related to cell type (self-interaction filters, $\boldsymbol{\beta}_i^{\text{self}}$), and how (normal distribution), but our algorithm can generalize to any choices for these. However, these choices, along with the choice to use a Gaussian approximation for variational inference, allow for a simpler, faster algorithm that can make use of off-the-shelf convex optimizers.

Overall, our method provides an unsupervised approach to the categorization of dynamical properties that differ among neurons. This approach complements well-established dynamical categories such as "Type I" and "Type II" neurons, as discussed in e.g. [16–18] that are established based on mathematical properties of their underlying differential equations models: specifically, the bifurcation that leads to a neuron's spiking. An interesting avenue for future work would be to compare these categories with what we find with the present approach.

Hierarchical generative models can easily be modified to incorporate multimodal data (as in [7]), so long as appropriate distributions for each modality can be specified. Thus our method could be used to identify multimodal cell types in terms of their transcriptomic and/or morphological properties as well as their functional ones, as an alternate approach to previous work ([19, 20]). Indeed, our approach can be easily applied to clustering problems outside of neuroscience, whenever there exists a cluster structure among individual entities, and each entity generates many samples of data, requiring only a change of how the data is described by a GLM.

That our approach provides the greatest performance improvements with data from many neurons may make it well-suited for application to *in vivo* recordings of brain activity. Modern recording technologies allow experimenters to simultaneously measure the activity of

hundreds to thousands of neurons at a time. However, the noise inherent to the data and the effect of unmeasured inputs on *in vivo* activity limit the accuracy of fitted parameters of single-cell neural dynamical models. In this work, our simultaneous method improves the accuracy of fitted parameters, suggesting that it may be able to overcome these challenges.

Applying this algorithm to *in vivo* neural recordings is an exciting avenue for future work. This would require expanding the framework to include connections between observed neurons and noisy inputs from unobserved ones. There has been much research on cell-type-specific connectivity suggesting that certain types are more likely to synapse on each other, and affect each other in stereotyped ways [21–25]. Including aspects of connectivity and/or noisy inputs with the cell-type-specific parameters $\boldsymbol{\beta}$ may thus lead to improved estimates of network effects on neural activity. For example, Jonas and Kording use a similar generative model and simultaneous approach that predicts connectivity between cells as well as data about each cell's position [7]. Although their method was applied to data about measured anatomical connectivity, such approaches could be modified to work with inferred connectivity instead. Given the long-term goal of identifying functional cell types from *in vivo* data, using this method to identify cell types according to inferred intrinsic electrophysiological parameters together with inferred connectivity parameters is a promising direction.

Such network models with functional cell types that such an algorithm may produce can complement the growing body of theoretical literature regarding such networks [26, 27]. Such work provides ideas about how to interpret cell-type-specific properties and interactions that our method may discover in the context of a neural circuit; in return, our method's discovery of cell-type structure in neural data would highlight the biological relevance of such theoretical work.

## Supporting information

**S1 Text. Supplementary information.** Additional details regarding methods and results, as well as supplementary analyses.
(PDF)

## Acknowledgments

We thank the Allen Institute founder, Paul G. Allen, for his vision, encouragement, and support.

## Author Contributions

**Conceptualization:** Daniel N. Zdeblick, Eric T. Shea-Brown, Daniela M. Witten, Michael A. Buice.

**Formal analysis:** Daniel N. Zdeblick.

**Funding acquisition:** Daniela M. Witten, Michael A. Buice.

**Methodology:** Daniel N. Zdeblick, Eric T. Shea-Brown, Daniela M. Witten, Michael A. Buice.

**Project administration:** Eric T. Shea-Brown, Daniela M. Witten, Michael A. Buice.

**Resources:** Michael A. Buice.

**Software:** Daniel N. Zdeblick.

**Supervision:** Eric T. Shea-Brown, Daniela M. Witten, Michael A. Buice.

**Validation:** Daniel N. Zdeblick.

**Visualization:** Daniel N. Zdeblick.

**Writing – original draft:** Daniel N. Zdeblick, Michael A. Buice.

**Writing – review & editing:** Daniel N. Zdeblick, Eric T. Shea-Brown, Daniela M. Witten, Michael A. Buice.

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
