## [Decision Letter · Decision Letter 0]

3 Apr 2023

Dear Zdeblick,

Thank you very much for submitting your manuscript "Modeling functional cell types in spike train data" for consideration at PLOS Computational Biology. As with all papers reviewed by the journal, your manuscript was reviewed by members of the editorial board and by several independent reviewers. The reviewers appreciated the attention to an important topic. Based on the reviews, we are likely to accept this manuscript for publication, providing that you modify the manuscript according to the review recommendations.

Sincerely,

Xue-Xin Wei

Academic Editor

PLOS Computational Biology

Marieke van Vugt

Section Editor

PLOS Computational Biology

Reviewer's Responses to Questions

**Comments to the Authors:**

Reviewer #1: The review is uploaded as an attachment

Reviewer #2: The authors propose a hierarchical statistical model, building on generalized linear point process models for single-neuron recordings, to identify functional cell type clusters in neural data. The paper demonstrates the utility and practicality of a hierarchical model to directly probe hypotheses of clustered functional types within a single step, rather than separate GLM and clustering models. The presentation is great: a tutorial-like to guide that could walk a broad audience in the neuro community through the methods in addition to the motivations and results. This manuscript was enjoyable to read and it would make a good addition to PloS-CB pretty much as is. I have included a few suggestions that the authors may wish to incorporate into their final version.

Comments

1. What was the total computation time to fit each approach for each example? This could depend a lot on the size of the grid of penalty terms (\\lamdba^{stim} and \\lambda^{self}) considered.

2. Figure 2B shows that the sequential method primarily fails to fit the larger negative refractory period terms. Additional interpretation here would be helpful. It appears that the ridge penalty is too strong for these large magnitude terms when applied to single neurons, each with independent penalty terms. Pointing out this over-shrinkage effect would provide a nice contrast to the high variance of the sequential method in the next section (Figure 4D).

3. Page 7, after equation 9, states “This hierarchical model has two hyperparameters, K, \\lambda^{stim}”. What about the spike history penalty term, \\lambda^{self}?

Reviewer #3: The authors present a method to estimate functional cell types by fitting a generative model that simultaneously estimates neurons' cluster identities and the functional (GLM) parameters per neuron (that depend on their cluster). Their method generally outperforms a sequential approach where GLMs are first fit per neuron and then those GLM coefficients are clustered. 

I found the paper to be thoroughly written and I only have minor comments. 

-I found equation 2 hard to follow. I think it could be useful to have a supplemental figure demonstrating how the covariates are constructed and fed into the GLM.

-I would clarify why you are clustering just based on the self-interaction filters in the main text. The results clearly are better when using only self-interaction filters - why do you think this is the case?

-I found the description of EV_ratio in the Methods hard to follow - how does an average (either the model average or PSTH average, which is staying the same) covary with individual trial responses?

-For figs 1-5, please put the panel letter in the upper left (which is what I always see) instead of lower left. I kept getting thrown off when looking at figures.

-It's hard to judge how realistic the simulated data is. To demonstrate that the method works on simulated data that is similar to neural data, I would recommend the following additional approach: 1) Fit the model to your actual data - this model will be your ground truth model; 2) Generate simulated data from that ground truth model (which should hopefully be fairly similar to the true neural data); 3) Fit a new model to the simulated data and see how well it can recover the ground truth parameters.

- For the line before section 3.2.1: I believe 0.6 should be 0.06

- I'm confused by the following statement: "Given that different populations of neurons are responsible for changes inANLL and EVratio, it is no surprise that these metrics scale differently with the numberof presentations used for training." Why does the number of training examples affect different populations of neurons (high and low firing) differently?

-In section 3.2.2, could you please expand upon (or make more concrete) the following sentences:  "...tend to have similar correlations with specific attributes (especially dendrite type). This suggeststhat these attributes may not be linked to cell type per se, but rather to anelectrophysiological feature that varies continuously between cell types." This result seems to be an important practical difference between the simultaneous and sequential approaches, and it can be challenging to understand upon the first read.

-I would mention the github repository in the paper so people know where to find it. I would also strongly recommend that the repository/code is complete and easy-to-follow (e.g. commented) prior to acceptance (so others can easily use and adapt your method).

**Have the authors made all data and (if applicable) computational code underlying the findings in their manuscript fully available?**

Reviewer #1: Yes

Reviewer #2: Yes

Reviewer #3: Yes

PLOS authors have the option to publish the peer review history of their article (what does this mean?). If published, this will include your full peer review and any attached files.

Reviewer #1: No

Reviewer #2: No

Reviewer #3: No

Figure Files:

Data Requirements:

Reproducibility:

References:

---

## [Decision Letter · Decision Letter 1]

12 Sep 2023

Dear Zdeblick,

We are pleased to inform you that your manuscript 'Modeling functional cell types in spike train data' has been provisionally accepted for publication in PLOS Computational Biology.

Best regards,

Xue-Xin Wei

Academic Editor

PLOS Computational Biology

Marieke van Vugt

Section Editor

PLOS Computational Biology

Reviewer's Responses to Questions

**Comments to the Authors:**

Reviewer #2: The revisions the authors have made have improved the clarity of the manuscript. My comments have all been answered. The study is well-motivated and presented, and I strongly recommend this paper for publication in PlosCB.

Page 12: should “Here we use oracle to select…” include an article before oracle?

Reviewer #3: Thank you for thoroughly addressing all concerns - I think this is a very well-done research study.

I have one minor suggestion:

-I found aspects of section 3.2.1 slightly hard to follow (in particular the 2nd paragraph of the section at the top of page 14), as you do many different analyses in the section. For instance, at the bottom of that 2nd paragraph, you mention Fig 5F, but it seems those analyses come in future paragraphs. Adding some additional clarifying text to this section would be helpful.

**Have the authors made all data and (if applicable) computational code underlying the findings in their manuscript fully available?**

Reviewer #2: Yes

Reviewer #3: Yes

PLOS authors have the option to publish the peer review history of their article (what does this mean?). If published, this will include your full peer review and any attached files.

Reviewer #2: No

Reviewer #3: No

---

## [Editor Report · Acceptance letter]

20 Sep 2023

PCOMPBIOL-D-23-00351R1 

Modeling functional cell types in spike train data

Dear Dr Zdeblick,

I am pleased to inform you that your manuscript has been formally accepted for publication in PLOS Computational Biology. Your manuscript is now with our production department and you will be notified of the publication date in due course.

With kind regards,

Anita Estes
